# Chemical Sensing Properties of BaF$_2$-Modified *h*BN Flakes towards Detection of Volatile Organic Compounds

**Boitumelo J. Matsoso** [1,2] 🆔, **Clara Garcia-Martinez** [3], **Thomas H. Mongwe** [4,5], **Bérangère Toury** [1], **José P. M. Serbena** [3,]* **and Catherine Journet** [1,]* 🆔

1. Laboratoire des Multimatériaux et Interfaces, UMR CNRS 5615, Univ-Lyon, Université Claude Bernard Lyon 1, CEDEX, F-69622 Villeurbanne, France; boijo.matsoso@gmail.com (B.J.M.); berangere.toury-pierre@univ-lyon1.fr (B.T.)
2. Department of Inorganic Chemistry, University of Chemistry and Technology Prague, Technická 5, Dejvice, 166 28 Prague, Czech Republic
3. Centro Politécnico, Departamento de Física, Universidade Federal do Parana, P.O. Box 19.44, Curitiba 80060-000, PR, Brazil; claraigarcia2@hotmail.com
4. DSI-NRF Centre of Excellence in Strong Materials, Johannesburg 2050, South Africa; nyathithomas@gmail.com
5. Molecular Science Institute, School of Chemistry, University of the Witwatersrand, Johannesburg 2050, South Africa
* Correspondence: serbena@fisica.ufpr.br (J.P.M.S.); catherine.journet@univ-lyon1.fr (C.J.)

**Abstract:** The application of BaF$_2$-modified *h*BN flakes as rapid response and recovery as well as sensitive chemoresistive sensing device materials for detection of acetone and/or ethanol is presented in this study. Modification of the *h*BN flakes was achieved by using the modified polymer derived ceramics (PDCs) process through the use of 0–10 wt% BaF$_2$ and 5 wt% Li$_3$N. Upon exposure to individual acetone and ethanol vapours, room temperature sensing studies revealed high LoD values ($-144$–460 ppm$_{acetone}$ and $-134$–543 ppm$_{ethanol}$) with extremely compromised sensitivities of $-0.042$–$0.72 \times 10^{-2}$ ppm$^{-1}$$_{acetone}$ and $-0.045$–$0.19 \times 10^{-2}$ ppm$^{-1}$$_{ethanol}$ for the structurally improved 5–10 wt% BaF$_2$-modified *h*BN flakes. Moreover, enhanced sensing for 0–2.5 wt% BaF$_2$-modified *h*BN flakes, as shown by the low LoDs ($-43$–86 ppm$_{acetone}$ and $-30$–62 ppm$_{ethanol}$) and the high sensitivities ($-1.8$–$2.1 \times 10^{-2}$ ppm$^{-1}$$_{acetone}$ and $-1.5$–$1.6 \times 10^{-2}$ ppm$^{-1}$$_{ethanol}$), was attributed to the presence of defects subsequently providing an abundance of adsorption sites. Overall, the study demonstrated the importance of structural properties of *h*BN flakes on their surface chemistry towards room temperature selective and sensitive detection of VOCs.

**Keywords:** *h*BN; acetone; ethanol; chemoresistive sensing; sensitivity; BaF$_2$-modification

## 1. Introduction

Recent rapid population growth and development of numerous industrial and technological sectors has led to increased emission of toxic and hazardous gases in the environment. Among these toxic and hazardous emissions, volatile organic compounds (VOCs) constitute the highest percentage owing to their significance as solvents and/or components in chemical, food, and healthcare industries. Acetone and ethanol are amongst the extensively used industrial VOCs despite their extreme flammability [1,2]. However, their fumes can be easily inhaled, and prolonged exposure to their vapours can result in serious health effects to humans and animals at large [3,4]. For instance, as the main non-methane organic pollutant, acetone concentrations higher than 173 ppm were reported to be responsible for severe central nervous system damage as well as causing headaches, fatigue, or narcosis [1]. As a result, threshold detection limit for any acetone gas sensor to be used for environmental safety and health purposes was set to 250 ppm, with an assumption of an 8 h exposure time weighted average [1,5,6]. Unlike acetone, the threshold detection limits of ethanol are relatively higher, being ≤1000 ppm over an 8 h exposure

time weighted average [4,6,7]. Despite ethanol being a simple aliphatic alcohol and having significant roles in pharmacology for drug dissolution, in chemical synthesis, as a solvent and/or constituent compound, and in the food industry, its cytotoxicity is of great concern. This is because ethanol concentration levels above its threshold detection limit were found to lead to continuous lachrymation and coughing [4,8]. As a result, development of portable gas sensor devices exhibiting fast response and high sensitivity towards low concentrations of acetone and/or ethanol vapours is of great importance, both for indoor and outdoor applications, and has recently gained great prominence.

Over the last few decades, the use of various active materials including semiconducting metal oxides and nitrides, polymers, ionic membranes, and salts has been extensively explored for chemoresistive gas sensing devices [9,10]. The most popular groups are the semiconducting metal oxides and nitrides, as they provide an array of advantages such as low-power-consumption, flexibility, as well as simple architecture [11–14]. However, their high operating temperatures (>250 °C) and extreme sensitivity to humidity [11–13,15] hinder their fundamental application as high-performance acetone and/or ethanol sensing devices. Recently, the successful development and implementation of graphene-based VOCs sensors brought growing attention towards other members of the layered two-dimensional (2D) materials family, such as transition metal dichalcogenides (TMDs) and hexagonal boron nitride (*h*BN) nanosheets [13,16–20]. More specifically, nanocrystalline *h*BN nanosheets offer a route for improving sensitivity of gas sensors due to their high thermoconductivity, mechanical strength, and chemical stability [21,22]. Additionally, *h*BN nanosheets have good adsorption properties due to the partially ionic B–N chemical bonds and their 2D nature, which subsequently lead to high surface/volume ratio and enable total exposure of their atoms to the adsorbing gas molecules [23,24], thus rendering them excellent candidates as active materials in gas-sensing technology. Theoretically and experimentally, gas-sensing performance of *h*BN nanosheets was predicted for $CH_4$, $F_2$, $NO_2$, $H_2$, $N_2$, $O_2$, and $CO_2$ [20,25]. However, to the best of our knowledge, their experimental performance towards VOCs is rather scarce. Therefore, the current study aimed to explore *h*BN flakes as promising materials for detecting acetone and ethanol as the commonly used industrial VOCs. Additionally, the study hopes to provide useful information on the influence of the physicochemical properties of *h*BN flakes upon screening with acetone and ethanol.

## 2. Materials and Methods

### 2.1. Material Synthesis and Sensor Preparation

The hexagonal boron nitride (*h*BN) nanosheets were synthesised based on the reported procedure, whereby borazine was used a source for boron and nitrogen atoms, lithium nitride ($Li_3N$, 99.4%, Alfa Aesar) [26,27], and barium fluoride ($BaF_2$, 99%, Alfa Aesar) [28]. Typically, pure borazine monomer was polymerized at 55 °C to generate a colourless liquid polyborazylene (PBN) [28,29]. Then, 5 wt% $Li_3N$ and 0–10 wt% $BaF_2$ [30] were added to PBN, and the suspension was homogenised for 10 min. The suspension was then heated to 200 °C for 1 h, followed by annealing of the solid-state polymer for 1 h at 1200 °C under inert nitrogen ($N_2$, 98%, Air Liquide, France) atmosphere [31,32]. The samples were then labelled as pristine *h*BN as well as 2.5 wt%, 5 wt%, and 10 wt% *h*BN for samples obtained after modification with 2.5, 5, and 10 wt% $BaF_2$. Finally, the pristine *h*BN and $BaF_2$-modified *h*BN samples were evaluated for chemical vapour sensing properties by screening their as-fabricated devices against the polar aprotic (acetone) and the polar protic (ethanol) analytes (Table S1). For device fabrication, dispersions of 2 mg/mL of the *h*BN samples in 4 mg/mL of hexadecyltrimethylammonium bromide (CTAB) were sonicated for 30 min at 60 °C, then at 0 °C for a further 30 min [32]. The solutions were then stored at 0 °C so as to facilitate precipitation of hydrated crystals [33]. After careful decantation of the supernatant, −100 μL of the *h*BN dispersions were drop-casted onto an FR4 substrate containing interdigitated electrodes (ENIG-Electroless Nickel Immersion Gold, Micropress S.A.; active area −64 mm$^2$) and dried in an oven at 100 °C for 30 min (Scheme 1) [34,35].

The *h*BN nanosheets were then screened with the polar aprotic (acetone) and polar protic analytes (ethanol) for their potential as chemical vapour sensors.

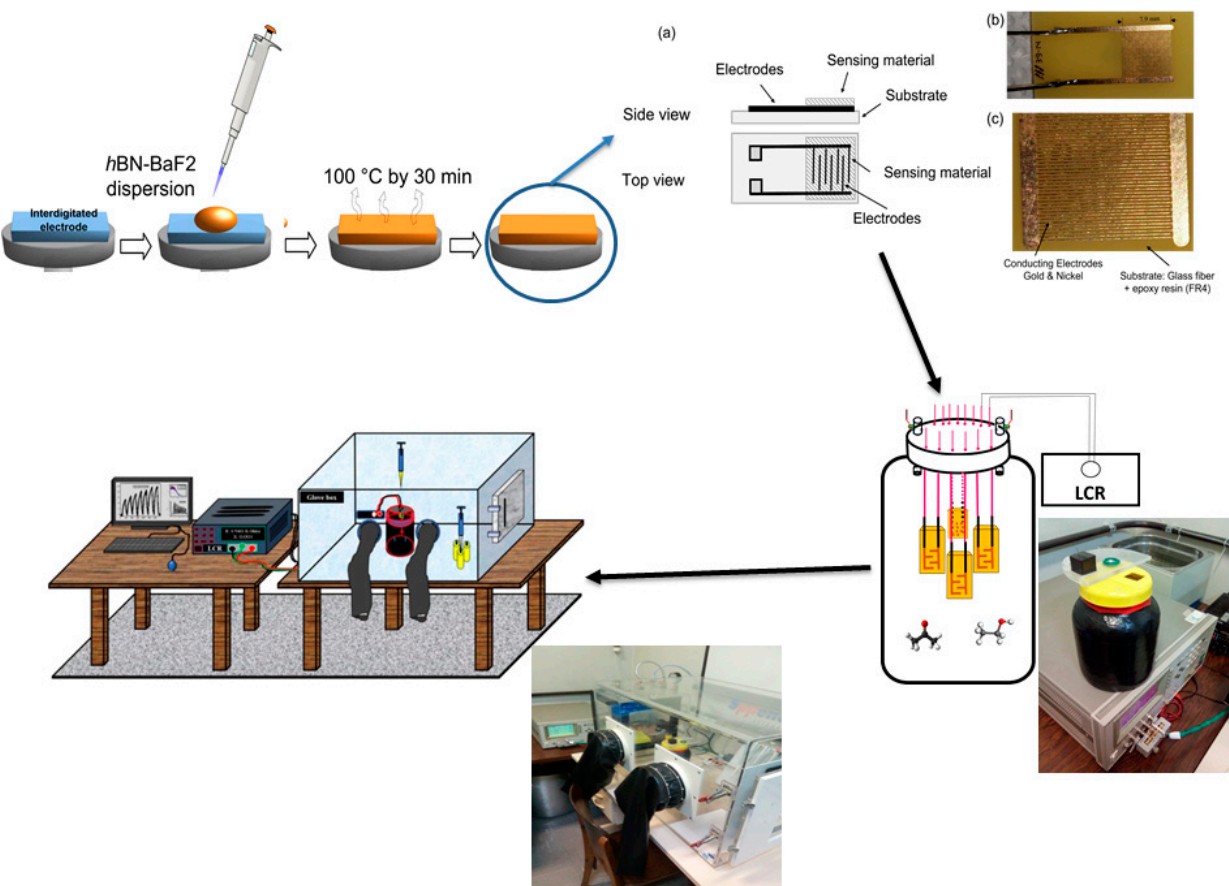

**Scheme 1.** Representation of the sensor preparation on the interdigitated electrodes (**a–c**) and the device measurements chamber.

### 2.2. Characterisation

Prior to device fabrication, the morphological determination of the *h*BN samples was evaluated using the MET Phillips CM120 transmission electron microscope (TEM) at 120 kV after the nanosheets were supported onto a holey carbon mesh on a Cu grid. Their degree of crystallisation was studied using the HORIBA Jobin-Yvon Labram Evolution Raman spectrometer at the wavelength of 532 nm. The surface areas of the samples were investigated from the BELsorpII mini after degassing for 4 h at 100 °C and measuring adsorption and desorption isotherms of ultra-pure $N_2$ gas at 77 K. The chemoresistive measurements were recorded on an LCR meter (Agilent 4284A), and experiments were conducted in a glovebox under dry $N_2$ atmosphere.

### 3. Results

#### 3.1. Structural Analysis

The morphology of the device active materials was investigated using transmission electron microscopy (TEM), which showed that the *h*BN samples were mainly overlapping, well-defined, and plate-like nanostructures of 0.89 ± 0.01, 2.9 ± 0.7, 3.3 ± 0.3, and 3.2 ± 0.7 μm in dimension for pristine *h*BN nanoflakes as well as the modified *h*BN samples after modification with 2.5, 5, and 10 wt% of $BaF_2$, respectively (Figure 1). The improvement in *h*BN nanosheets size upon addition of $BaF_2$ to the pre-ceramization mixture can be attributed to the faster melting of $Li_3N$. This was facilitated by $BaF_2$ [28], consequently leading to improved crystallisation of *h*BN from PBN. As such, the effect of the addition

of BaF$_2$ to crystallinity and textual properties of the *h*BN nanosheets was studied using Raman spectroscopy and BET analysis.

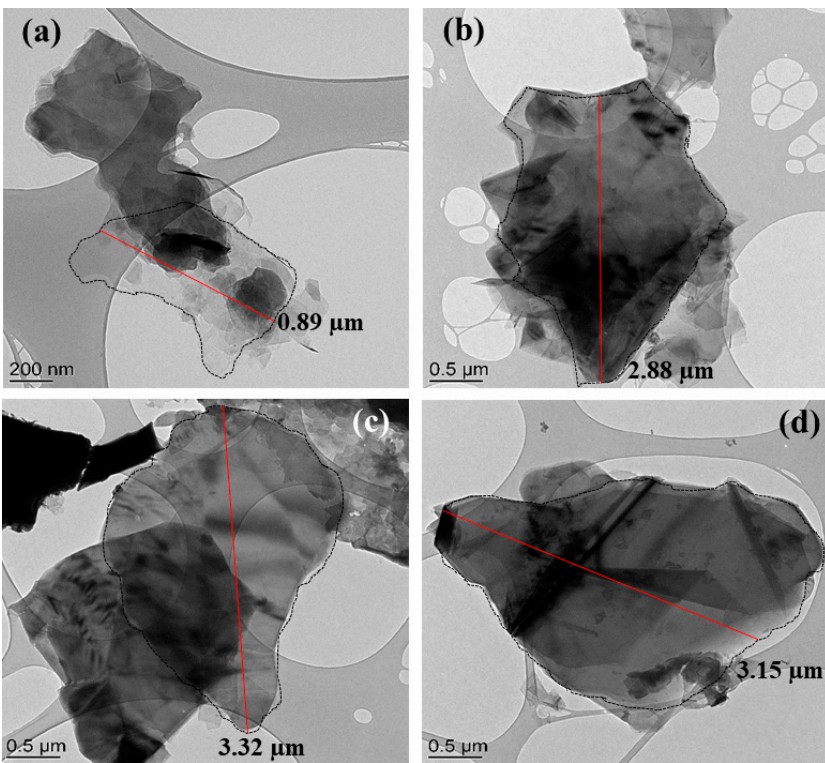

**Figure 1.** TEM images of (**a**) pristine *h*BN electrode materials and *h*BN samples after modification with (**b**) 2.5, (**c**) 5, and (**d**) 10 wt% BaF2 [28].

The crystalline structural properties of the samples by Raman spectroscopy revealed formation of *h*BN nanostructures as evidenced by the presence of the first-order active Raman vibrating mode of *h*BN (E$_{2g}$) centred at $-1365.4 \pm 1.6$ cm$^{-1}$ (Figure 2a) [28,36]. Additionally, correlation of the finite-size effects within *h*BN with the inherent broadening of the Raman vibrational modes (Figure 2a, inset) showed the dependence of crystallinity of the *h*BN nanosheets to the addition of BaF$_2$. For instance, the bandwidths (FWHM) values decreased from 17.01 cm$^{-1}$ for the pristine *h*BN sample to 11.07 cm$^{-1}$ for the 5 wt% BaF$_2$- modified *h*BN sample, followed by a slight increase to 11.9 cm$^{-1}$ recorded for the 10 wt% BaF$_2$- modified *h*BN sample. The narrowing of the bandwidths can be ascribed to formation of larger crystallites, as observed from the TEM micrographs (Figure 1c), as well as subsequent improvement in crystallinity and quality of the *h*BN nanosheets. As the *h*BN nanosheets are to be used as the active material in chemical sensor devices, determination of their surface areas and pore size distributions is very crucial in assessing their potential application within the field of sensor technology. Based on the multi-point Brunauer–Emmet–Teller (BET) method shown in Figure 2b, the samples exhibited a type II isotherm—an indication of the formation of macroporous or non-porous materials. As a result, the specific surface area was then determined to be 8.7 m$^2$/g for the pristine *h*BN sample, whilst modification with 2.5, 5, and 10 wt% of BaF$_2$ led to surface areas of 3.5, 3.6, and 2.9 m$^2$/g, respectively [28]. The decreasing surface area was expectedly due to the formation of larger crystalline planer flakes, thus limiting the amount of adsorption sites for N$_2$ and resulting in adsorption and desorption of the monolayer of N$_2$ predominantly on the material's external rough surface on the basal planes. However, the results still indicate that the samples had sufficient adsorption sites for the chemical vapours, thus exhibiting potential for application in the chemical sensor technology.

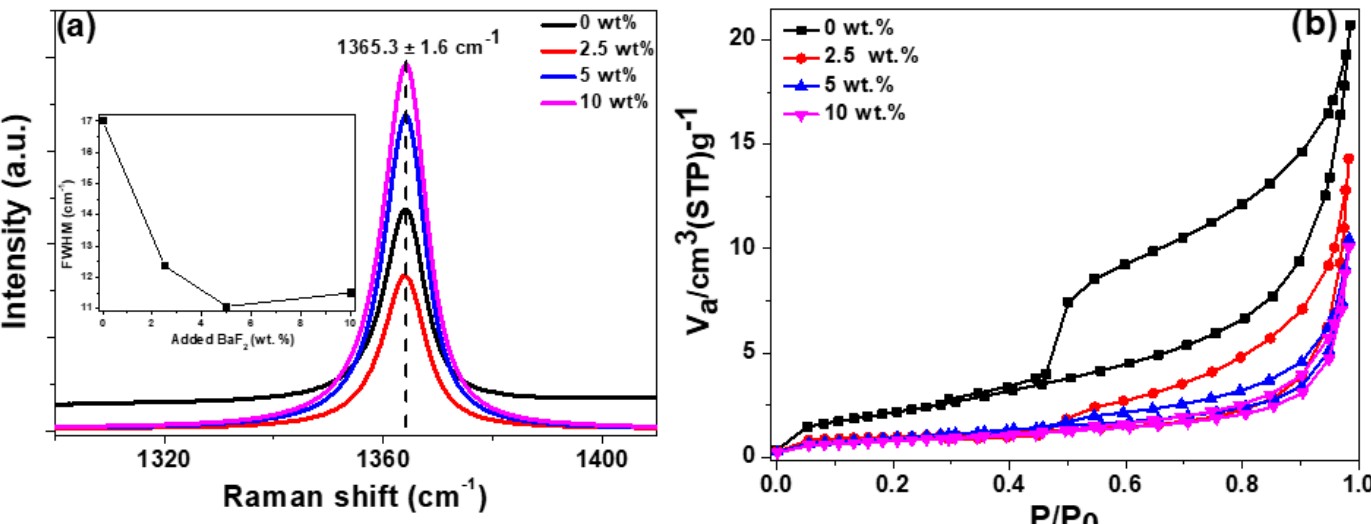

**Figure 2.** (**a**) Raman plots and (**b**) adsorption isotherms of pristine and 2.5, 5, and 10 wt% BaF$_2$-modified *h*BN samples. Inset: bandwidth of the Raman spectra as a function of BaF$_2$ wt% [28].

### 3.2. Sensing Properties of the hBN Nanosheets

#### 3.2.1. Resistance of the Sensors

The electrical resistance of the active materials as a function of the frequency with increasing concentration of an individual or a mixture of analytes was plotted so as to determine to the optimum operating frequency for future experiments. From Figure 3 and Figure S1, great dependence of the sensor response to the structural properties of the *h*BN nanosheets was seen upon exposure to increasing concentrations of acetone and/or ethanol vapours. For instance, the rapid drop in resistance with increasing frequency for all samples can be ascribed to the presence of defects within the layered 2D structure of the *h*BN nanosheets [37]. Regardless of whether the nanosheets exhibited a well-defined morphology, as in the case of the 5 wt% BaF$_2$-modified *h*BN sample, or were defective, as seen with the pristine and the 2.5 wt% BaF$_2$-modified *h*BN samples, the presence of defects influenced the resistance of the resultant sensing device. We applied an alternating electric field due to the movement of charge carriers over relatively long distances so as to overcome the defects as well as the potential barriers of the layered 2D nanostructure [38]. As such, the best operating frequencies for the active materials in the modified *h*BN-based devices for acetone and/or ethanol were determined to be in the range of 1–3 kHz. The selection of the optimum operating frequency of the sensors was such that the LoD should have been lower whilst the sensitivity was higher (Figures S2–S6). The operating frequency was found to be better than and/or comparable to that of the commonly used active materials for detection of acetone and/or ethanol vapours, especially for room temperature-based sensors. For instance, Mutuma et al. reported operating frequencies of 3–10 kHz for differently nitrogen-doped hollow carbon sphere-based acetone sensors [39]. Moreover, Li et.al reported that zinc oxide-based nanosheets functioned best at the operating frequency of 3 kHz for detection of acetone, although the devices had to be heated to a temperature of 280 °C [40]. Generally, the results showed that the morphology of the *h*BN sensor material has a great influence on the sensing capability towards either acetone or ethanol vapours.

#### 3.2.2. Performance of the Sensors

Based on the optimum operating frequency for each *h*BN-based sensor (Table 1), the performance of the devices was determined from their responses upon exposure to the vapours and the recoveries after removal of the analyte vapour. Figure 4 shows that all active materials exhibited excellent gas-sensing performance for individual acetone or ethanol vapours, as shown by response and recovery times, which were <100 s (Table S2) at optimum frequencies and exposure of 160 ppm analyte vapour. However, exposure to the

vapour mixture showed compromised response times for all samples except for the sensor devices based on the 5 wt% BaF$_2$-modified $h$BN samples (t$_{res}$ −56 s and t$_{rec}$ −15 s). This could be attributed to the poor interaction of both acetone and ethanol molecules on the basal planes of $h$BN flakes. Interestingly, for the majority of samples, the results indicated that the samples behaved as $p$-type semiconducting devices, shown by the increase in sensor resistance upon exposure to the analyte vapour and a depletion in sensor resistance on removal. On the contrary, the sensor performance for the devices fabricated out of 0 wt% BaF$_2$-modified $h$BN samples exhibited $n$-type semiconducting behaviour (Figure 4i), an indication of the prolonged depletion of the pre-generated holes as well as increased amount of hopping sites due to the defective nature of the 0 wt% BaF$_2$-modified $h$BN sample, thereby leading to increased conductivity upon exposure to the vapour mixture.

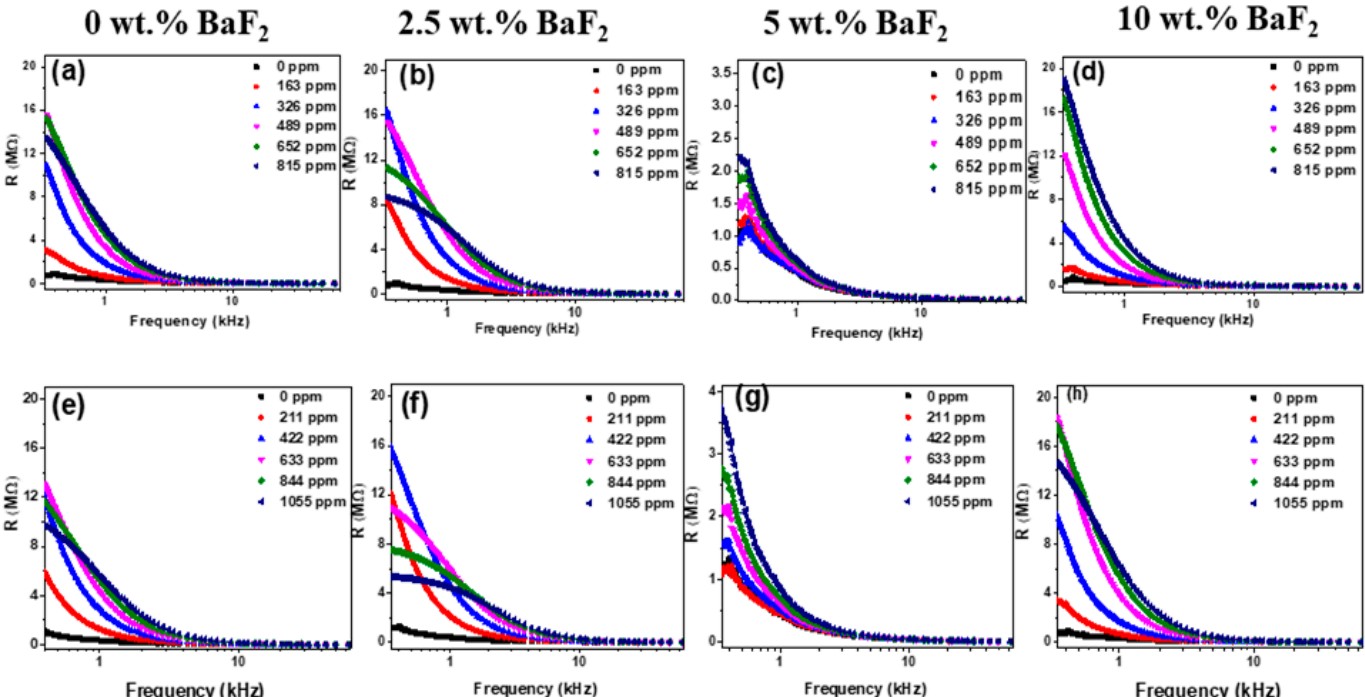

**Figure 3.** Dependence of sensor resistance on frequency with increasing concentrations of (**a–d**) acetone and (**e–h**) ethanol vapours.

**Table 1.** Optimum frequency ($f$), determined limit of detection concentration (LoD), and sensitivity ($S$) for detection of acetone, ethanol, and acetone: ethanol mixture.

| Analyte | Parameter | 0 wt% | 2.5 wt% | 5 wt% | 10 wt% |
|---|---|---|---|---|---|
| **Acetone** | $f$ (kHz) | 1 | 1 | 3 | 3 |
| | $S$ ($\times 10^{-2}$ ppm$^{-1}$) | 1.8 | 2.1 | 0.042 | 0.72 |
| | LoD (ppm) | 86.2 | 43.2 | 460 | 144 |
| **Ethanol** | $f$ (kHz) | 1 | 3 | 3 | 1 |
| | $S$ ($\times 10^{-2}$ ppm$^{-1}$) | 1.5 | 1.6 | 0.045 | 0.19 |
| | LoD (ppm) | 30.4 | 61.7 | 542.6 | 133.5 |
| **Acetone: Ethanol (1:1)** | $f$ (kHz) | 1 | 3 | 1 | 1 |
| | $S$ ($\times 10^{-3}$ ppm$^{-1}$) | −9.1 | 7.0 | 7.2 | 10.0 |
| | LoD (ppm) | 30.4 | 18 | 197 | 439 |

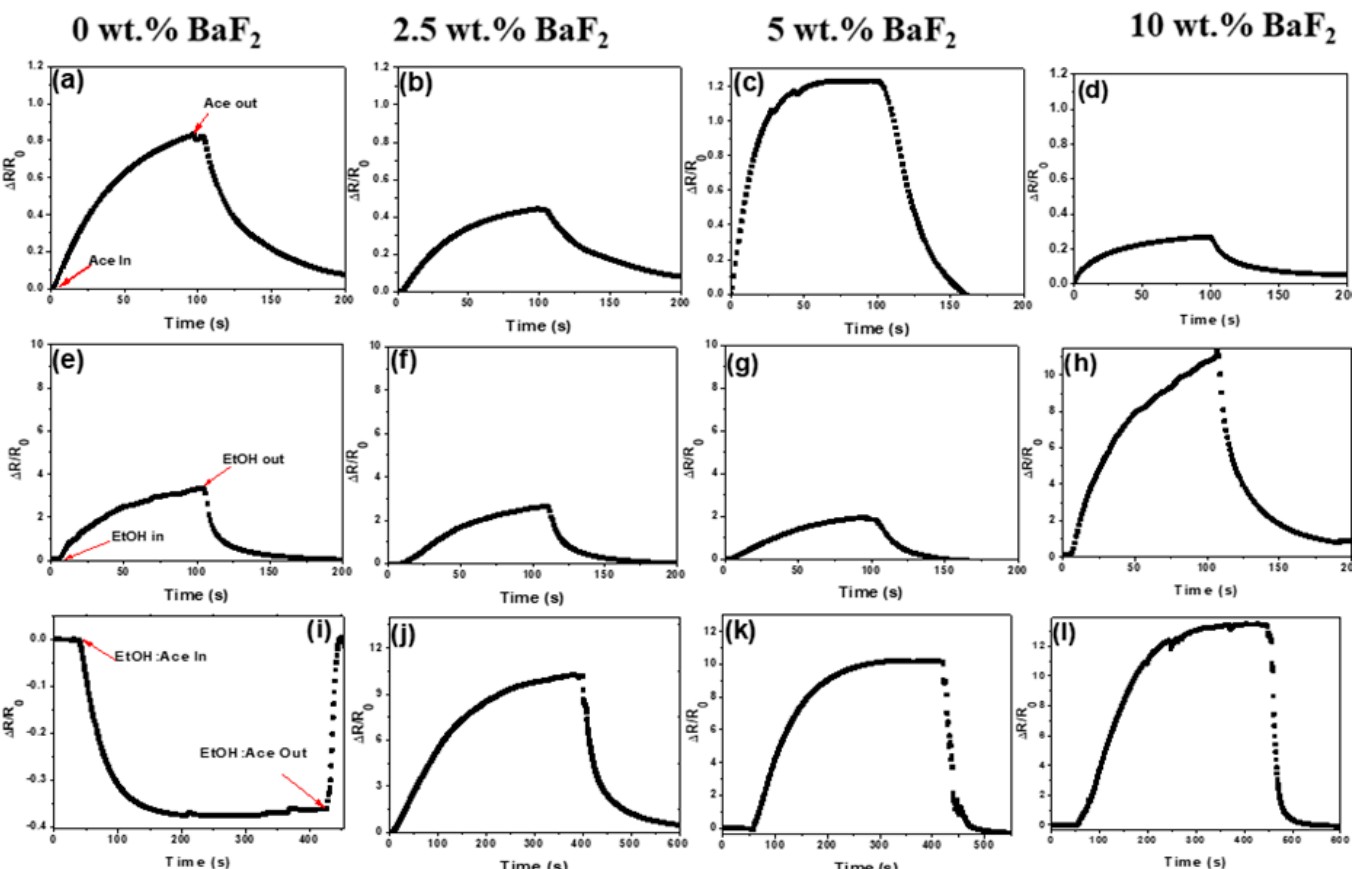

**Figure 4.** Responses and recovery times for devices fabricated after exposure and removal of 160 ppm of (**a–d**) acetone, (**e–f**) ethanol, and (**i–l**) acetone: ethanol vapours at optimum operating frequencies.

Furthermore, better performance was recorded for the *h*BN nanosheets that exhibited well-defined nanosheets morphology and structural properties (Figure 4c,d). For instance, response times less than 60 s were recorded for the 5 wt% and the 10 wt% BaF$_2$-modified *h*BN samples upon exposure to acetone vapour, whilst recovery times <50 s were observed for both samples after removal of acetone (Table S3). The observations can be ascribed to the abundance of active surface adsorption sites on the basal planes of *h*BN nanosheets through the generation of more ionised adsorbed oxygen species (i.e., $O_2^-$, $O^-$, or $O^{2-}$, Scheme 2a,b) [41]. These adsorbed oxygen functional groups enabled faster conversion of acetone to carbon dioxide (Scheme 2(ci)) facilitated by the weak interaction of acetone molecules to the *h*BN surface. Furthermore, an improved saturation platform was observed for the 5 wt% BaF$_2$-modified *h*BN sample (Figure 4c) in comparison with other *h*BN-based devices. This could have been due to the presence of larger basal planes for the 5 wt% BaF$_2$-modified *h*BN sample, which enabled adsorption and coverage of a larger surface by the acetone molecules, thus leading to the observed faster response and recovery times for this sample. Owing to the large basal plane, faster response (56 s) and recovery (15 s) times were recorded upon exposure of the 5 wt% BaF$_2$-modified *h*BN-based sensor device to the mixture of acetone and ethanol. Moreover, this could be suggestive of the selectivity of this *h*BN-based sensor towards acetone in the presence of other VOCs.

On the other hand, the longer response times for pristine (−74 s) and 2.5 wt% BaF$_2$-modified (−61 s) *h*BN samples (Table S3) towards detection of acetone could be attributed to the retention of charge carriers as a result of prolonged electron-hopping effect generated by the structural defects. Subsequently, it can be suggested that these defects led to strong interaction of acetone molecules with the *h*BN surface, thereby resulting in longer recovery times of −95 s and −71 s for the pristine and the 2.5 wt% BaF$_2$-modified *h*BN samples, respectively. The retention of charge by defects was also shown by the extremely long

responses ($-113$ s) and recoveries ($-75$ s), thus translating to poor sensing performance of the 2.5 wt% $BaF_2$-modified *h*BN samples for detection and selectivity of VOCs. In the presence of a mixture of gases, roles of various surface structural properties were shown by the varying response and recovery times (Figure 4i–l). Expectedly, great sensing performance was exhibited by the 5 wt% $BaF_2$ *h*BN-based device, as shown by the short response ($-56$ s) and recovery ($-75$ s) times. This indicated that the basal plane of the structurally improved 5 wt% $BaF_2$ *h*BN nanoflakes (Figure 4k) enhanced the charge transfer to both analytes. On the other hand, the defective-prone 2.5 wt% $BaF_2$ *h*BN (Figure 4j) nanostructures suffered a somewhat compromised sensing performance, as indicated by the extremely long response ($-113$ s) and recovery ($-75$ s) times. Interestingly, different sensing behaviour was shown by the variously modified *h*BN devices, with the pristine *h*BN device showing the n-type semiconducting behaviour (Figure 4i), whereas all the $BaF_2$-modified *h*BN devices displayed a *p*-type semiconducting behaviour (Figure 4j–l). The results show the significance of the structural properties in the manner of which each *h*BN active material transfer charge carried upon exposure to a mixture of ethanol and acetone vapours.

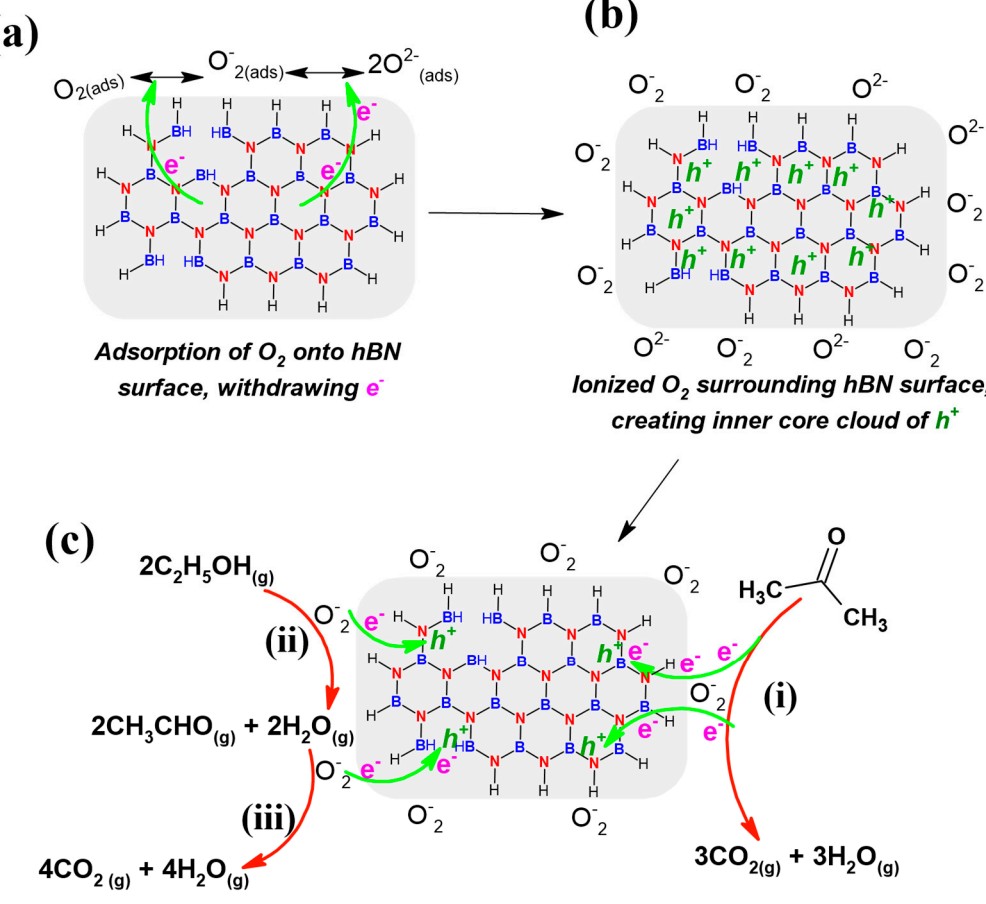

**Scheme 2.** Adsorption and reaction mechanism of acetone or ethanol on the *h*BN surface.

Regarding the performance of the sensors on exposure to ethanol vapour, the devices showed poor response times ($-70$–$80$ s, Table S3), an indication of poor transfer of charge carriers of the *h*BN active materials to the adsorbed ethanol molecules, therefore leading to a prolonged electron-hopping effect. Albeit, the sensing performance towards ethanol and/or acetone is governed mostly by the interaction of ethanol molecules with the gener-

ated adsorbed $O_2^-$, $O^-$, or $O^{2-}$ species [27,42], the long response times during exposure to ethanol for all devices are suggestive of the difficult absorption of the more polar ethanol molecules on the surface and the grain boundaries of $h$BN nanosheets in comparison with the adsorption mechanism of acetone. Moreover, owing to the two-step reduction process of ethanol on the active materials (Scheme 2(cii,ciii)) [27,42], this could have contributed to the observed longer response times of ethanol in all samples. In this case, when the sensing device was exposed to ethanol vapour, the reduced gas molecules were first oxidised into an acetaldehyde molecule (Scheme 2(cii)), which then required more oxygen anions for the final conversion into carbon dioxide and water (Scheme 2(ciii)). However, the shorter recovery times of −34 s, −56 s, −34 s, and −40 s for pristine, 2.5 wt%, 5 wt%, and 10 wt% $BaF_2$-modified $h$BN-based devices (Table S3), respectively, are suggestive of the weak interaction of the ethanol molecule with the basal planar $h$BN surface. In hindsight, the results show that layered $h$BN nanosheets with well-defined morphologies and improved properties can have excellent functionality as good resistance-based VOCs sensors, similar to their zero-bandgap graphene, small band-gap semiconducting transitional metal dichalcogenides (TMDs), and conducting mxenes counter-parts [16,43,44].

### 3.2.3. Determination of Sensing Parameters

At the chosen optimum operating frequencies, sensing parameters of each $h$BN-based device, given by the concentration limit of detection (LoD, Equation (S2)) and the sensitivity ($S$, Equation (S3)), were estimated from the plots of sensor resistance and/or response ($\Delta R/R_0$, Equation (S4)) against the concentration of analyte of interest (Figure 5 and Figures S2–S4). As seen in Figure 5, the sensor responses were increasing with either acetone or ethanol vapour concentration, an indication that the $h$BN-based nanosheets in this study were $p$-type semiconducting in nature. However, the structural morphology of the $h$BN nanosheets was found to play a profound role in the overall sensing performance of the devices. For instance, very low sensitivity and extremely high LoD values were determined for the devices based on the $h$BN nanosheets exhibiting improved structural properties. In particular, $4.2 \times 10^{-4}$ ppm$^{-1}$ and 460 ppm (Table 1) were estimated as the sensitivity and the LoD values for the sensing device based on 5 wt% $BaF_2$-modified $h$BN nanosheets for acetone detection, whilst ethanol detection registered $4.5 \times 10^{-4}$ ppm$^{-1}$ and 543 ppm, respectively. Similarly, sensitivity and LoD values were estimated for the sensing device based on 10 wt% $BaF_2$-modified $h$BN nanosheets towards the detection of both acetone and ethanol (Table 1). Despite the 5 wt% $BaF_2$-modified $h$BN-based devices displaying fast response and recovery times upon exposure to acetone and ethanol (Figure 4c,g), their overall poor sensing performance could be attributed to the weak interactions between the carbonyl groups on the analyte molecules with the $h$BN basal surface. A similar argument could also be used for the observed LoD (144 ppm$_{acetone}$ and 134 ppm$_{ethanol}$) and sensitivity ($7.2 \times 10^{-3}$ ppm$^{-1}_{acetone}$ and $1.9 \times 10^{-3}$ ppm$^{-1}_{ethanol}$) values for the 10 wt% $BaF_2$-modified $h$BN-based devices on exposure to increasing concentrations of both acetone and ethanol vapours. Interestingly, slight improvement in the sensing performance of the $h$BN nanosheets exhibiting improved structural properties was observed upon exposure to the mixture of ethanol and acetone (Table 1 and Figure S2c,d). For instance, low LoD values of 197 ppm and 439 ppm were estimated for 5 wt% and 10 wt% $BaF_2$-modified $h$BN-based devices, whereas higher sensitivities were determined to be $7.0 \times 10^{-3}$ ppm$^{-1}$ and $1.0 \times 10^{-2}$ ppm$^{-1}$, respectively. This is suggestive of enhanced transfer of charge carriers. Moreover, the defects were observed to have little impact on the sensing performances of 0 wt% and 2.5 wt% $BaF_2$-modified $h$BN-based devices, as high sensitivity and low LoD values were estimated upon detection of a VOCs vapour mixture of ethanol and acetone. In spite of the retention of charge carriers by defective sites, these acted as capture sites for the adsorption of a larger volume of analyte molecules, thus corresponding to low LoD values of 30.4 ppm and 18 ppm for 0 wt% and 2.5 wt% $BaF_2$-modified $h$BN-based devices, respectively.

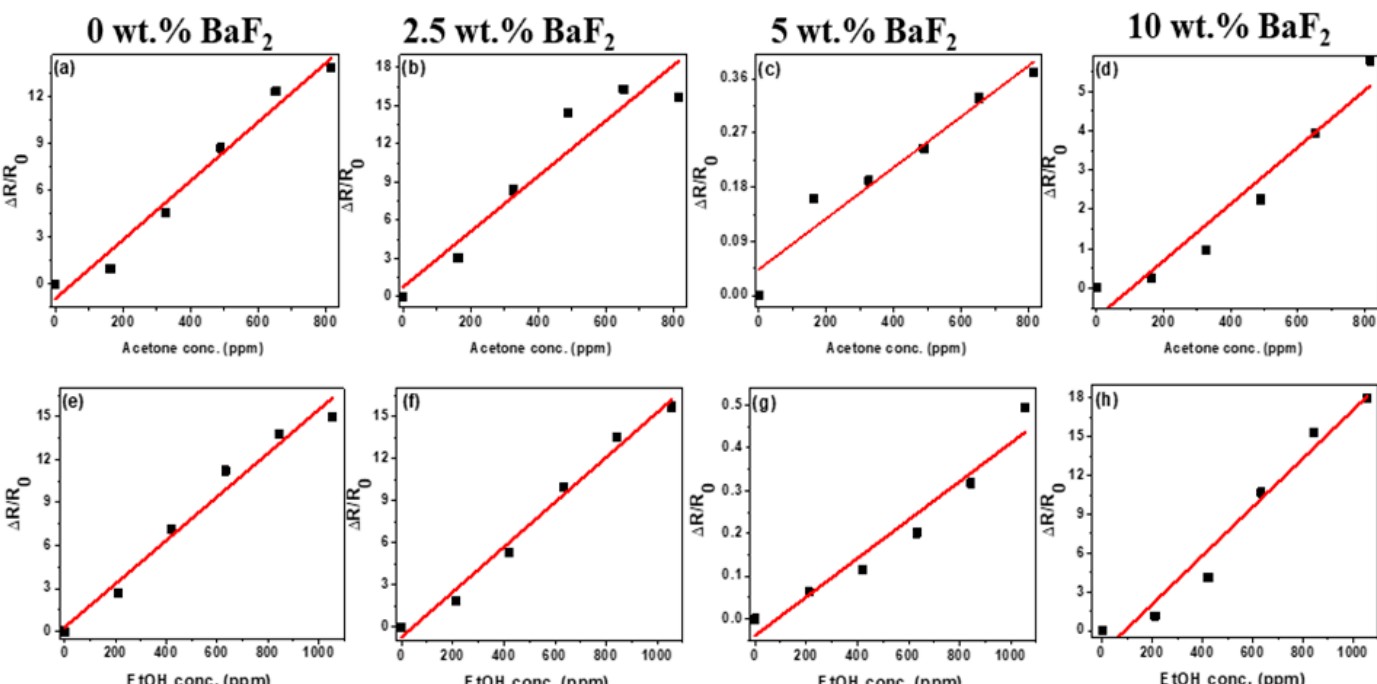

**Figure 5.** Sensor response versus concentrations of (**a**–**d**) acetone and (**e**–**h**) ethanol at optimum operating frequencies.

On the other hand, pristine and defective 2.5 wt% BaF$_2$-modified *h*BN samples registered lower concentration limits of detection and high sensitivity values (Table 1). The lower values are indicative of improved sensing performance, which could be ascribed to the stronger surface interaction of the analyte molecules with the *h*BN nanosheets. Furthermore, the abundant nitrogen and/or boron vacancies (V$_N$ or V$_B$) in the defective structure of *h*BN could have acted as electron capture sites upon exposure to either acetone or ethanol [45,46], thereby resulting in improved sensitivity. Interestingly, regardless of the theoretical studies showing that the even, smooth surfaces of *h*BN nanosheets enabled full access of their atoms to the adsorbing gas molecules, thus allowing for larger sensor area per unit volume and improving the sensitivity [25–27], our results showed that the formation of well-defined morphology *h*BN nanosheets greatly hindered their performance towards the detection of volatile organic compounds. In summary, the results indicated that our synthesised *h*BN nanosheets are potential candidates for room temperature chemoresistive gas sensors (Table 2), as the obtained values were comparable to those of graphene, conductive metal oxides, mxenes, as well as transition metal dichalcogenides (TMDs) [25]. Moreover, the sensors were found to be stable even after 18 months of storage as approximately 75%, 50%, 92%, and 8% decreases in sensor responses were observed for 0 wt%, 2.5 wt%, 5 wt%, and 10 wt% modified *h*BN samples after exposure to 160 ppm of acetone (Figure S7). Likewise, the effect of the structural properties was observed by the sensitivity values, which were recorded after 18 months and were found to be 0.04, 0.1, 0.04, and 0.12 as well as 0.4, 2.3, 0.06, and 0.2 $\times$ 10$^{-2}$ ppm$^{-1}$ against acetone and ethanol for 0 wt%, 2.5 wt%, 5 wt%, and 10 wt% modified *h*BN samples, respectively (Table S4).The 5 wt%-BaF$_2$ *h*BN samples produced reproducible results in comparison with other active materials even after 18 months of storage, and this can be attribute to the improved crystallinity of the nanoflakes, which thus limited and/or slowed the degradation of the *h*BN nanostructures.

Our reported sensor performance, specifically for the pristine and the 2.5 wt% BaF$_2$ modified *h*BN samples towards acetone or ethanol, showed better and/or relatively comparable LoD values to those of their 2D counterparts such as graphene oxide, transition metal dichalcogenides (TMDs), and mxenes (Table 2). For instance, the pristine and the 2.5 wt% BaF$_2$ modified *h*BN-based sensors showed significantly lower LoD values than sensors based on TMDs and metal oxides for acetone and ethanol molecules; however, they

were still relatively higher than those of graphene-and mxene-based sensors [16,46–55]. Better LoDs in mxene- and graphene-based sensors were attributed to the abundance of oxygenated and/or ligand functional groups, which ultimately facilitated an improved adsorption–desorption mechanism of the VOC molecules. On the other hand, improvement in structural properties upon addition of BaF$_2$ to the preceramic mixture of *h*BN, especially 5 wt%, greatly compromised their sensing performance towards VOCs. This was ascribed to the lack of defect sites, which subsequently led to fewer oxygenated functional groups, thereby affecting the sensing properties of the 5–10 wt% BaF$_2$ modified *h*BN-based sensors. The important point in this work is the low LoD value for the defective *h*BN samples towards detection of both VOCs compared to other reports in literature. Generally, the results indicate that, with controlled improvement in the structural properties of *h*BN, enhanced sensing performance of *h*BN nanosheets towards both protic and aprotic VOCs can be achieved.

**Table 2.** Comparative sensor data for the *h*BN samples with other related materials for VOCs sensors.

| Analyte | Active Material | Sensor Type | Conc. Range (ppm) | Temp. (°C) | LoD (ppm) | Ref |
|---|---|---|---|---|---|---|
| **Acetone** | ZnO | Chemoresistive | 5–1000 | 300 | 10 | [40] |
| | ZnO/Gr | | 10–10,000 | 280 | 13.3 | [50] |
| | pristine *h*BN 2.5wt% BaF$_2$-*h*BN 5wt% BaF$_2$-*h*BN 10wt% BaF$_2$-*h*BN | | 0–100 | RT | 86 43 460 144 | This work |
| | α-Fe$_2$O$_3$/rGO | | 5–500 | 225 | 5 | [53] |
| | g-C$_3$N$_4$/WO$_3$ | | N/A | 340 | 100 | [43] |
| | Ti$_3$C$_2$T$_x$ | Electrochemical | N/A | RT | 0.05 | [42] |
| **Ethanol** | SnO$_2$/MoS$_2$ | Chemoresistive | N/A | 280 | 50 | [54] |
| | ZnO/GO | | 10–1000 | 400 | 10 | [55] |
| | pristine *h*BN 2.5wt% BaF$_2$-*h*BN 5wt% BaF$_2$-*h*BN 10wt% BaF$_2$-*h*BN | | 0–100 | RT | 30 62 543 134 | This work |
| | Ti$_3$C$_2$T$_x$ | Electrochemical | N/A | RT | 0.10 | [42] |

MoS$_2$ → molybdenum disulphide, ZnO → zinc oxide, *h*BN → hexagonal boron nitride, BaF$_2$ → barium fluoride, rGO → reduced graphene oxide, T$_3$C$_2$T$_x$ → bimetallic Mxene, SnO$_2$ → tin oxide, GO →graphene oxide, WO$_3$ → tungsten oxide, gC$_3$N$_4$ → graphitic carbon nitride, α-Fe$_2$O$_3$ → alpha ferric oxide.

## 4. Conclusions

Following the modification of hexagonal boron nitride (*h*BN) flakes produced through addition of barium fluoride (BaF$_2$) in a polymer-derived ceramics (PDCs) synthesis technique, their sensing performance towards volatile organic compounds (VOCs) was then investigated. The results showed that the sensing devices fabricated from the *h*BN nanostructures were active for acetone or ethanol detection, and their sensing performance was dependent on the structural properties of the nanostructures. For instance, the pristine and the 2.5 wt% BaF$_2$ modified *h*BN-based sensors exhibited improved sensing performance towards both analytes. This was shown by the low LoDs ($-43$–86 ppm$_{acetone}$ and $-30$–62 ppm$_{ethanol}$) and the high sensitivities ($-1.8$–$2.1 \times 10^{-2}$ ppm$^{-1}$$_{acetone}$ and $-1.5$–$1.6 \times 10^{-2}$ ppm$^{-1}$$_{ethanol}$), which can be attributed to the defective domains on these samples, which subsequently provided an abundance of adsorption sites for the analyte molecules of interest. Despite the rapid response and recovery times for the 5–10 wt% BaF$_2$ modified *h*BN-based sensors, the improved 2D morphology for the *h*BN flakes was found to hinder the good sensing performance of *h*BN nanostructures towards acetone and ethanol. This was highlighted by the high LoD values of $-144$–460 ppm and $-134$–543 ppm, with extremely poor sensitivities of $-0.042$–$0.72 \times 10^{-2}$ ppm$^{-1}$ and $-0.045$–$0.19 \times 10^{-2}$ ppm$^{-1}$

for acetone and ethanol for the structurally improved 5–10 wt% BaF$_2$-modified *h*BN flakes, respectively. The loss in sensor activity was also attributed to the relatively low surface areas of the *h*BN nanostructures (2.9 to 3.5 m$^2$/g). Overall, the study demonstrated that the structural properties of the *h*BN flakes should be taken into serious consideration prior to the design of room temperature sensing devices for VOCs.

**Supplementary Materials:** The following are available online at https://www.mdpi.com/article/10.3390/chemosensors9090263/s1, Figure S1: Dependence of sensor resistance on frequency with increasing EtOH: Acetone concentrations for (a) 0, (b) 2.5, (c) 5, and (d) 10 wt% BaF$_2$-modified *h*BN based sensors, Figure S2: Sensor response versus EtOH: acetone concentrations (a) 0, (b) 2.5, (c) 5, and (d) 10 wt% BaF$_2$-modified *h*BN based sensors at optimum operating frequencies, Figure S3: Sensor resistance as a function of analyte concentration for (a-d) acetone, (e-h) ethanol, and (i-l) ethanol: acetone; red line indicates the estimated LoD resistance of the corresponding sensor at the optimum frequency, Figure S4: The sensitivities of sensors data based on *h*BN-BaF$_2$ (a) 0 wt%, (b) 2.5 wt%, (c) 5 wt%, and (d) 10 wt% for acetone as a function of frequency; dashed line indicates the optimum operating frequency, Figure S5: The sensitivities of sensors data based on *h*BN-BaF$_2$ (a) 0 wt%, (b) 2.5 wt%, (c) 5 wt%, and (d) 10 wt% for ethanol as a function of frequency; dashed line indicates the optimum operating frequency, Figure S6: The sensitivities of sensors data based on *h*BN-BaF$_2$ (a) 0 wt%, (b) 2.5 wt%, (c) 5 wt%, and (d) 10 wt% for ethanol:acetone as a function of frequency; dashed line indicates the optimum operating frequency, Figure S7: Responses and recovery times for (a) 0 wt%, (b) 2.5 wt%, (c) 5 wt%, and (d) 10 wt% BaF$_2$-modified *h*BN devices fabricated 18 months ago and after exposure and removal of 160 ppm of acetone at optimum operating frequencies, Table S1: Properties of the studied analytes, Table S2: Optimisation parameters for the *h*BN sensors, Table S3: Response and recovery times at 2 mg·mL$^{-1}$ concentration of the *h*BN dispersion, Table S4: Reproducibility of the *h*BN sensors.

**Author Contributions:** Conceptualisation, B.J.M., C.J., B.T. and J.P.M.S.; methodology, B.T. and J.P.M.S.; validation, B.J.M., C.J., B.T. and J.P.M.S.; formal analysis, B.J.M., C.G.-M.; investigation, B.J.M., C.G.-M., T.H.M.; resources, C.J., B.T. and J.P.M.S.; data curation, B.J.M., C.G.-M., T.H.M.; writing—original draft preparation, B.J.M.; writing—review and editing, B.J.M., C.J., B.T., J.P.M.S.; supervision, C.J., B.T., J.P.M.S.; project administration, C.J., B.T. and J.P.M.S.; funding acquisition, C.J., B.T., J.P.M.S. All authors have read and agreed to the published version of the manuscript.

**Funding:** This work was financially supported by European Graphene flagship (Grant number: 785219), CNRS Délégation Rhône Auvergne, and Université Claude Bernard Lyon 1. This study was financed in part by the Coordenação de Aperfeiçoamento de Pessoal de Nível Superior–Brasil (CAPES)–Finance Code 001 and the Conselho Nacional de Desenvolvimento Científico e Tecnológico–Brasil (CNPq).

**Institutional Review Board Statement:** Not applicable.

**Informed Consent Statement:** Not applicable.

**Data Availability Statement:** Not applicable.

**Acknowledgments:** Sincere gratitude to the Group of Organic Optoelectronic Devices (GOOD) at the Federal University of Parana (UFPR-Curitiba, Brazil) for technical assistance and support during chemical sensing experiments. The authors thank the 'Centre Technologique des Microstructures', CTµ, (Université Lyon 1) for providing access to TEM and SEM facilities, CECOMO for access to Raman spectroscopy, and Science et Surface (Ecully, France) for XPS analyses.

**Conflicts of Interest:** The authors declare no conflict of research and financial interest.

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
