# Peer review of "Chemical Sensing Properties of BaF2-Modified hBN Flakes towards Detection of Volatile Organic Compounds"

_chemosensors, doi:10.3390/chemosensors9090263_

Round 1

Reviewer 1 Report

Bellow the author could find my comments:

  1. A sketch of the sensor layout as well as the fabrication process, as appropriate, should be included.
  2. Materials and Methods section - gas sensing tests - more details of the sensing setup are required, e.g. how was concentration of the target gas adjusted in the sensing setup? What was the volume of the test chamber? What was the gas flow rate? A schematic diagram or photo of the gas sensing test setup should be included.
  3. Throughout the manuscript the Authors state about the morphological changes of BaF2-modified hBN flakes with additional BaF2 concentration. However, they do not state the actual thickness of the performed films. The film thickness needs to be measured and reported.
  4. The description of the figure caption does not correlate with the content. In the manuscript twice presented Figure 2 and Table 1.
  5. The data reported in Figure 2 (page 6) are meaningless. How the authors can claim that the optimum operating frequency of the sensor devices based on 2.5 wt% and wt5% BaF2-modified hBN are 1 and 3 kHz, respectively? The presented data do not demonstrate clear evidence of optimum operating frequency. Moreover, please, use the range of vertical axis (Y-axis) the same for all measured data.
  6. In Figure 4 please use the same range of X-axis and Y-axis for all presented data.
  7. What about stability and robustness of BaF2-modified hNB under relative humidity? I strongly suggest to measure sensor device at different humidity levels, i.e. 30-80% RH.
  8. Reproducibility of the device needs to be discussed given its natural importance.
  9. Throughout the manuscript the Authors state about the selectivity to acetone at room temperature. The selective sensing should be claimed only in the case that the tests would have been done in gas mixtures. However, this was not the case. Please remove the statement of selective detection.

Author Response

Reviewer 1

The authors proposed a volatile organic compounds sensor based the hexagonal boron nitride nanosheets. It is very interesting to the VOCs detection.  

1.Page 9, line 300,"their overall poor sensing performance could be attributed to the weak interactions between the carbonyl groups on the analyte molecules with the hBN basal surface." was mentioned in the paper. The reason needs to be explained in detail.

As the sensing mechanism between the VOCs and the hBN surface is through the hydrogen bonding between the analyses’ carbonyl groups and defective sites on the hBN active material, then higher sensitivities and large LoD values could be indicative of the need for adsorption of a large quantity of analyte molecules to facilitate the transfer of charge carriers from hBN to the VOCs. As such, this can then be ascribed to the weak interaction between the molecule and the active material.

2.Page 11, Can the concentration range of the sensor studied only be 0-100ppm? If so, please explain the reason. If not, please supplement the experiment of other concentration range.

Working concentration range of the sensors is limited by the LoD and LoQ parameters. Therefore, for good quantification of the analyte as well as qualitative analyte detection, the choice of the concentration range is governed by the LoQ value which is lower than 100 ppm.

Table S2: Optimization parameters for the hBN sensors

Analyte

Parameter

0 wt%

2.5 wt%

5 wt%

10 wt%

Acetone

f (kHz)

1

1

3

3

LoQ (ppm)

184

132

-

239

Ethanol

f (kHz)

1

3

3

1

LoQ (ppm)

135

92

-

230

3.Figure 2 of the paper should be Figure 3. Some of the pictures in the paper are small, and the author needs to provide clearer original pictures. For example, figure 2, figure 3, figure 4 and figure 5.

We apologize for the typo with regard to the figure labeling. Also, larger pictures have been provided for figures 2-5.

4.There is a problem with the typesetting of the paper, and there are redundant blanks on many pages. For example, pages 4, 5, 6, 9.

The redundant blanks in the manuscript were due to the formatting the original Word document to the template of the journal. We have adjusted the indentation.

Reviewer 2 Report

The present study investigates the VOCs sensing properties of
BaF2-modified hBN flakes. It is found that BaF2 addition on the surface
of hBN could improve the VOCs detection to ethanol and acetone, which
attributes to defect and abundant adsorption sites. The paper is
well-written and therefore can be accepted after clarifying the issues
listed below :

1. Nitrides are well-known for their chemical stability at harsh
temperatures (doi.org/10.1007/s12613-020-2143-8) Author may mention this interesting aspect in the introduction

2. How selective hBN is towards other VOCs and non-VOCs gas?

3. Please confirm the stability of the hBN sensor, either by long-term
exposure, or structural confirmation by XRD, SEM/TEM after gas sensing
measurement.

Reviewer 3 Report

The authors proposed a volatile organic compounds sensor based the hexagonal boron nitride nanosheets. It is very interesting to the VOCs detection.  

  1. Page 9, line 300,"their overall poor sensing performance could be attributed to the weak interactions between the carbonyl groups on the analyte molecules with the hBN basal surface." was mentioned in the paper. The reason needs to be explained in detail.
  2. Page 11, Can the concentration range of the sensor studied only be 0-100ppm? If so, please explain the reason. If not, please supplement the experiment of other concentration range.
  3. Figure 2 of the paper should be Figure 3. Some of the pictures in the paper are small, and the author needs to provide clearer original pictures. For example, figure 2, figure 3, figure 4 and figure 5.
  4. There is a problem with the typesetting of the paper, and there are redundant blanks on many pages. For example, pages 4, 5, 6, 9.

Round 2

Reviewer 1 Report

Thank you for adding in the manuscript a sketch of the sensor layout and sensor preparation procedure. However, I still have some comments and suggestions:

  1. In Figure 4, I suggest to use the same Y axis which can help better distinguish the difference of the sensor responses between 0, 2.5, 5 and 10 wt.% BaF2. For example, for acetone (4a-4d), dR/R0 in the range of 0 – 1.2. Please apply similar strategy to all sensors, i.e. 4e-4l.
  2. Regarding the comments about Relative humidity (#7), you claim that the measurements were done under 30% RH. Unfortunately, I did not find any information about it throughout the manuscript. Moreover, in comment #2 you wrote that the measurements were provided under dry N2 atmosphere.

         I understand that the study on effect of relative humidity were not performed, however I strongly recommend you to carry out those measurements for this work.

  1. In the manuscript there is no discussion regarding the results in Figure 4 (i-l), where the authors used gas mixtures of acetone and ethanol (1:1).

Author Response

Kindly see attachment
